# Jingmen Tick Virus in Ticks from Kenya

**DOI:** 10.3390/v14051041

**Published:** 2022-05-13

**Authors:** Edwin O. Ogola, Anne Kopp, Armanda D. S. Bastos, Inga Slothouwer, Marco Marklewitz, Dorcus Omoga, Gilbert Rotich, Caroline Getugi, Rosemary Sang, Baldwyn Torto, Sandra Junglen, David P. Tchouassi

**Affiliations:** 1International Centre of Insect Physiology and Ecology, Nairobi P.O. Box 30772-00100, Kenya; eogola@icipe.org (E.O.O.); domoga@icipe.org (D.O.); grotich@icipe.org (G.R.); cgetugi@icipe.org (C.G.); rsang@icipe.org (R.S.); btorto@icipe.org (B.T.); 2Department of Zoology and Entomology, University of Pretoria, Private Bag 20, Pretoria 0028, South Africa; adbastos@zoology.up.ac.za; 3Berlin Institute of Health, Institute of Virology, Charité—Universitätsmedizin Berlin, Corporate Member of Free University Berlin, Humboldt-University Berlin, Chariteplatz 1, 10117 Berlin, Germany; anne.kopp@charite.de (A.K.); inga.slothouwer@charite.de (I.S.); m.marklewitz@gmx.de (M.M.); 4German Centre for Infection Research (DZIF), Associated Partner Site Charité, 10117 Berlin, Germany; 5Kenya Medical Research Institute (KEMRI), Off Raila Odinga Way, Nairobi P.O. Box 54840-00200, Kenya

**Keywords:** arbovirus surveillance, JMTV, segmented genome, tortoise, ticks, human febrile illness

## Abstract

Jingmen tick virus (JMTV) is an arbovirus with a multisegmented genome related to those of unsegmented flaviviruses. The virus first described in *Rhipicephalus microplus* ticks collected in Jingmen city (Hubei Province, China) in 2010 is associated with febrile illness in humans. Since then, the geographic range has expanded to include Trinidad and Tobago, Brazil, and Uganda. However, the ecology of JMTV remains poorly described in Africa. We screened adult ticks (*n* = 4550, 718 pools) for JMTV infection by reverse transcription polymerase chain reaction (RT-PCR). Ticks were collected from cattle (*n* = 859, 18.88%), goats (*n* = 2070, 45.49%), sheep (*n* = 1574, 34.59%), and free-ranging tortoises (Leopard tortoise, *Stigmochelys pardalis*) (*n* = 47, 1.03%) in two Kenyan pastoralist-dominated areas (Baringo and Kajiado counties) with a history of undiagnosed febrile human illness. Surprisingly, ticks collected from goats (0.3%, 95% confidence interval (CI) 0.1–0.5), sheep (1.8%, 95% CI 1.2–2.5), and tortoise (74.5%, 95% CI 60.9–85.4, were found infected with JMTV, but ticks collected from cattle were all negative. JMTV ribonucleic acid (RNA) was also detected in blood from tortoises (66.7%, 95% CI 16.1–97.7). Intragenetic distance of JMTV sequences originating from tortoise-associated ticks was greater than that of sheep-associated ticks. Phylogenetic analyses of seven complete-coding genome sequences generated from tortoise-associated ticks formed a monophyletic clade within JMTV strains from other countries. In summary, our findings confirm the circulation of JMTV in ticks in Kenya. Further epidemiological surveys are needed to assess the potential public health impact of JMTV in Kenya.

## 1. Introduction

Jingmen tick virus (JMTV) is an emerging human pathogen first described in *Rhipicephalus microplus* ticks in Jingmen city (Hubei Province, China) in 2010 [1,2]. The public health importance of the virus has been demonstrated following its detection in patients presenting with mild to severe disease, and in ticks collected from the patients [2]. Jingmen tick virus was implicated in fatal human cases associated with Crimean–Congo hemorrhagic fever virus (CCHFV) in Kosovo in 2019 [3]. Variants of the virus have been detected in non-human primates in Uganda [4]. In Finland, Russia, and China, a JMTV-like virus known as Alongshan virus (ALSV) was recovered in patients with febrile illness and in ticks [5,6,7]. JMTV has also been detected in mosquitoes and in vertebrates including cattle and rodents in China [1,8,9]. However, the ecology of JMTV remains poorly described in Africa.

JMTV and related viruses have so far not been classified by the International Committee on Taxonomy of Viruses (ICTV) and are referred to as jingmenviruses [10,11]. Jingmenviruses form two phylogenetic groups. The viruses of one group are transmitted by blood-feeding arthropods and are vertebrate pathogenic (arbo-jingmenviruses) such as JMTV, Mogiana tick virus (MGTV), and Yanggou tick virus (YGTV) [2,10,11,12]. The second group contains viruses that only infect arthropods (arthropod-specific jingmenviruses) exemplified by the Guaico Culex virus (GCXV), Wuhan aphid virus (WHAV), and Shuangao insect virus (SHIV) groups [8,10,12]. Jingmenviruses have a genome comprising four segments of positive-sense single-stranded RNA and are distantly related to flaviviruses [1,10,11,12]. JMTV has a multisegmented genome comprising three monocistronic segments (segments 1, 2, and 3) and two separate open reading frames (ORF) in one segment (segment 4) [1,10,12]. The non-structural proteins of segment 1 (NSP1) and segment 3 (NSP2) seem to be homologous to those of the genus *flavivirus* (NS5, RNA-dependent RNA polymerase (RdRP) and NS3, helicase/protease) [1,10,12]. The close phylogenetic relationship of these two proteins provides an unusual evolutionary link between the non-segmented flaviviruses and the segmented jingmenviruses.

To date, there is no information regarding the presence of JMTV infections in vertebrates or ticks in Kenya. Towards predictive surveillance for arboviral threats, this study sought to assess the presence and distribution of JMTV in adult ticks in Kenya. Ticks collected from livestock hosts such as cattle, goats, and sheep and free-ranging tortoises from two pastoralist-predominant areas, Baringo and Kajiado counties were assessed.

## 2. Material and Methods

### 2.1. Ethical Approval

All experimental procedures were approved by the Kenya Medical Research Institute Scientific and Ethics Review Unit (KEMRI-SERU) (SERU protocol number 3312). Local authorities and village elders were consulted before the study was initiated and during each sampling visit. Additionally, informed oral consent was obtained from owners and/or caretakers before sampling domestic animals. Tick sampling from livestock was performed in accordance with ethical guidelines.

### 2.2. Study Sites, Tick Sampling and Morphological Identification

Adult ticks were collected from seven sites in two counties, Baringo and Kajiado, within the Great Rift Valley Region of Kenya. These counties (Figure 1) were selected because they have previous history of arboviral infections [13,14,15,16]. Four sampling sites were included in Baringo County, namely Ntepes, Sandai, Logumgum, and Kapkuikui, and three sites in Kajiado County, namely Oloisinyai, Oldorko, and Soweto (Figure 1). Baringo and Kajiado counties are home to Ruko and Olkirimatian conservancies, respectively, and have a semi-arid ecology. Both counties are inhabited by nomadic pastoralist communities that keep livestock (sheep, cattle, and goats). Both areas have a history of undiagnosed febrile human illness with poorly described aetiologies.

Adult ticks were collected from cattle (*Bos taurus*), goats (*Capra hircus*), and sheep (*Ovis aries*), as well as from tortoises (Leopard tortoise, *Stigmochelys pardalis*) in Baringo only, between August 2019 and July 2020 after the rains when arboviral activities are expected to be high [18]. Individual livestock specimens (cattle, goats, and sheep) were restrained to allow manual picking of attached adult ticks from the animal’s skin. The ticks were initially taken to a temporary field laboratory and cryopreserved in liquid nitrogen for transportation and subsequent storage in −80 °C freezers in the laboratory at *icipe* in Nairobi. Following surface sterilization with 70% ethanol and washing with deionized water to remove foreign particles from animals, the collected ticks were identified morphologically to species using established morphologic keys [19,20,21,22]. The ticks were pooled according to species, host, sex, and sampling sites in groups of up to eight individuals and stored in −80 °C freezers until further processing. Ticks that could not be identified to the species level were not pooled and analyzed individually.

### 2.3. Tortoise Sampling

Tortoise sampling was carried out in September 2019 at Kapkuikui, Baringo county after tick collection from livestock on a single occasion on August 2019. Free-ranging tortoises were physically restrained to allow blood collection from either the dorsal coccygeal vein or brachial sinus by a team including a registered animal health technician in accordance with the animal ethics protocol approved by KEMRI-SERU. About 0.5 mL of whole blood was collected aseptically using 2 mL BD Vacutainer with Ethylenediaminetetraacetic acid (EDTA) (Becton Dickinson, Sunnyvale, CA, USA) or approximately 50 µL whole blood spotted on FTA Classic Card Whatman (Merck KGaA; Darmstadt, Germany). The samples on 2 mL BD Vacutainer were cryopreserved in liquid nitrogen for transportation while the samples on FTA cards were stored at room temperature during transportation. All blood samples were stored at −80 °C during transfer to the laboratory at *icipe* in Nairobi, until further processing.

### 2.4. Tick Homogenization and Molecular Identification

Before genomic DNA extraction, ticks were frozen in liquid nitrogen and mechanically homogenized in 1.5 mL microcentrifuge tubes with zirconia beads (2.0 mm and 0.1 mm diameter) for 45 s in Mini-Beadbeater-16 (Biospec, Bartlesville, OK, USA). One mL DPBS (Dulbeccos phosphate-buffered saline) was added to each tick homogenate prior to centrifugation at 2500 revolutions per minute (rpm) at 4 °C for 10 min in a bench top centrifuge (Eppendorf 5430 R). Following the manufacturer’s protocol, genomic DNA was extracted from the pellet of *Amblyomma* ticks (that could not be identified at the species level by morphology) using the ISOLATE II Genomic DNA extraction kit (Bioline, London, UK). The supernatant was preserved at −80 °C for RNA extraction and virus isolation. Three PCR assays targeting the 16S ribosomal ribonucleic acid (rRNA), internal transcribed spacer 2 (ITS2), and cytochrome oxidase 1 (*CO1*) genes were employed for amplification (Appendix A) using the thermal cycling conditions described earlier [23,24,25]. The PCR products were examined on an ethidium bromide-stained 1.5% agarose gel and amplicons of the correct size purified for Sanger sequencing (Macrogen, Amsterdam, The Netherlands) using ExoSAP-IT (USB Corporation, Cleveland, OH, USA).

### 2.5. RNA Extraction and PCR Screening

Total RNA was extracted from 140 µL of tick homogenates (homogenization described above) and from tortoise whole blood using the QIAamp Viral RNA Mini Kit (Qiagen, Hilden, Germany) following the manufacturer’s instructions. The tick homogenates included both samples homogenized in pools and those homogenized individually. The remaining homogenates were preserved at −80 °C for virus isolation. Reverse transcription was carried out with 5 µL of the isolated RNA using the High Capacity cDNA Reverse Transcription (RT) kit (LifeTechnologies, Carlsbad, CA, USA) and 600 μM non-ribosomal random primers [26]. Samples were screened for JMTV infection by pan-JMTV RT-PCR using primers targeting the NS5 gene (Appendix A). PCRs were set up in a final volume of 25 µL containing 2.5 units of Mytaq DNA polymerase (Bioline, London, UK). The touch down thermal cycling conditions involved initial denaturation for 3 min at 95 °C, followed by 10 cycles of 20 s at 95 °C, 30 s at 64–56 °C and 30 s at 72 °C, and 35 cycles of 20 s at 95 °C, 30 s at 56 °C, and 30 s at 72 °C. The final extension lasted 5 min at 72 °C. The post-PCR analysis and Sanger sequencing were carried out as described above.

### 2.6. Virus Isolation and Quantification of Viral Genome Copies

Virus isolation attempts from JMTV-positive tick homogenates was performed in Vero E6 (*Ceropithecus aethiops*) and C6/36 (*Aedes*
*albopictus*) cells as previously described following filtration and antibiotic treatment approaches [27]. Briefly, before inoculation, cell culture medium was removed from the cells seeded in a 48-well plate (Nunc, Roskilde, Denmark) and 150 µL of cell line-specific medium without additives added. In the filtered approach, 100 µL of each sample (supernatant of tick homogenate) was passed through 0.45 µm sterile membrane filters (Merck Millipore, Billerica, MA, USA) using a syringe and 50 µL inoculated onto each cell line. Following inoculation, cells were allowed to adsorb for an hour and maintained in 300 µL media. Gibco Dulbecco’s modified Eagle’s medium (DMEM), used for maintaining Vero E6 cells, was supplemented with 5% fetal calf serum (FCS; Gibco, Thermo Fisher Scientific, Darmstadt, Germany) and 1% l-glutamine for Vero E6 cells. Gibco Leibovitz’s L-15 medium (L-15), used for maintaining C6/36 cells, was supplemented with 5% FCS. The antibiotic approach involved inoculating 50 µL of unfiltered tick homogenate and the addition of 100 U/mL penicillin together with 100 ug/mL amphotericin B. Vero E6 cells were incubated in a 5% CO_2_ atmosphere at 37 °C while C6/36 cells were incubated at 28 °C without CO_2_. The cells were monitored regularly for cytopathic effects (CPE) and four blind passages on fresh cells were carried out after every seven days.

During each passage, 75 µL of cell culture supernatant was collected for viral RNA isolation in the MagNa Pure 96 extraction system (Roche Diagnostics, Rotkreuz, Switzerland) followed by cDNA synthesis using Superscript IV reverse transcriptase and random hexamer primers (Invitrogen, Karlsruhe, Germany) as described by the manufacturer. A quantitative TaqMan Real-time PCR (qRT-PCR) using JMTV-specific primers and probe (Appendix A) was established to measure the amount of viral genome copies in the collected cell culture supernatants.

### 2.7. Library Preparation and Next-Generation Sequencing

Viral RNA was extracted from JMTV-positive tick homogenates using QIAamp Viral RNA Mini Kit (Qiagen, Hilden, Germany) as described above without adding carrier RNA. Viral RNA was further extracted from infectious cell culture supernatant using the NucleoSpin^®^ RNA Virus Kit (Macherey-Nagel, Düren, Germany) following the manufacturer’s protocol. The extracted RNA was used to sequence JMTV full genomes on an Illumina MiSeq Next-generation sequencing (NGS) platform as described earlier with modification [28] in that DNA libraries were prepared using the KAPA HyperPlus kit (Roche Diagnostics, Rotkreuz, Switzerland). After de-multiplexing and extracting the raw data in fastq format, the resulting paired reads were quality trimmed and filtered to remove Illumina adapters using BBDuk (filter = 27, trimk = 30; http://jgi.doe.gov/data-and-tools/bb-tools/, accessed on 25 January 2022). The read coverage was then normalized, employing the kmer-based normalization toolBBNorm with a target option of 40 and minimum depth of 6. After trimming and normalization, the duplicate reads were binned using Dedupe. Viral sequence identification involved mapping paired reads against JMTV segments S1, S2, S3, and S4 as well as by de novo assembly using Spades v3.11.1 (http://cab.spbu.ru/software/spades/, accessed on 25 January 2022). Genome analysis was performed using Geneious Prime, and nucleotides (nt) and deduced amino acids (aa) sequences were queried against the GenBank database using blastn and blastx searches [29,30].

### 2.8. Phylogenetic Analysis and Genome Characterization

The partial sequences (ITS2 and 16S rRNA) for ticks and fragments of NS5 genes for JMTV were analyzed in Geneious Prime and queried in GenBank [30]. Using the L-INS-i algorithm implemented in MAFFT, nucleotide sequences generated in this study were aligned with related sequences and the alignment was edited manually to adjust regions aligned ambiguously [29,31]. Maximum likelihood (ML) phylogenetic trees were estimated using PhyML v. 2.2.4 with the best fit model determined by Modeltest implemented in MEGA-X version 10.2.5 with nodal support being assessed through 1000 bootstrap replications [32].

Full JMTV genomes were analyzed using Geneious prime [30]. In summary, InterProScan implemented in Geneious prime was used to predict the location of transmembrane domains and signal peptides [29,33]. The location of potential N-glycosylation sites in the four segments were determined in the NetNGlyc v1.0 server (http://www.cbs.dtu.dk/services/NetNGlyc/, accessed on 25 January 2022). JMTV phylogenetic trees were inferred from MAFFT alignment using PhyML v. 2.2.4 with General-time-reversible (GTR) substitution models employing 1000 bootstrap replicates. Trees were midpoint rooted.

### 2.9. Statistical Analysis

The relative abundance of tick species collected from different hosts was estimated binomially and evaluated by Chi square tests at 95% confidence intervals using R version 4.1.2 [34]. To estimate JMTV prevalence in individual tick species, a frequentist model was used to implement maximum likelihood (ML) analysis in an online platform, EpiTools epidemiological calculator (http://epitools.ausvet.com.au/, accessed on 4 November 2021) [24,35,36,37].

### 2.10. Sequence Accession Numbers

The ITS2 and 16S rRNA gene fragments sequences were deposited in GenBank under the accession numbers ON212401–ON212405 and ON220154–ON220159, respectively. The JMTV NS5 gene sequences were deposited in GenBank under the accession numbers ON158817–ON158867. JMTV coding complete genome sequences were deposited in GenBank under the accession numbers ON186499–ON186526 (Appendix A).

## 3. Results

### 3.1. Tick Collection from Livestock and Tortoises

A total of 4550 ticks were collected from different hosts including cattle, goats, sheep, and tortoises from the study sites (Figure 1). The ticks comprised thirteen species in three genera including *Rhipicephalus (Rh.) appendiculatus*, *Rh. evertsi evertsi*, *Rh. pulchellus*, *Hyalomma*
*(Hy.) marginatum*, *Hy. rufipes*, *Hy. truncatum*, *Hy. impeltatum*, *Hy. albiparmatum*, *Amblyomma*
*(Am.) gemma*, *Am. nuttalli*, *Am. sparsum*, *Am. Variegatum* and *Am. lepidum* (Appendix A). Thirty-five *Amblyomma* ticks collected from tortoises could not be identified at the species level by morphology nor by sequencing of ITS2 and 16S rRNA gene fragments. These ticks are denoted as *Am.* sp., henceforth. The generated sequences showed 99%–100% pairwise nucleotide identities within the 16S rRNA gene fragment, and 98%–100% pairwise nucleotide identities within the ITS2 fragment. Amplification of the *CO1* gene was not successful. Phylogenetic analysis of ITS2 fragment sequences (1008 bp nucleotides in length) revealed that the *Am.* sp. ticks are most closely related to *Am. marmoreum* and *Am. loculosum*, showing pairwise nucleotide identities of 93%–94% to both species (Appendix A). Analysis of the 16S rRNA gene fragment (402 bp nucleotides in length) showed a similar relationship of the collected *Am.* sp. ticks to *Am. marmoreum* (93%–94% nt similarity). These data suggest that the collected *Am.* sp. ticks are distinct from formally recognized *Amblyomma* tick species for which homologous 16S rRNA and ITS2 data are available (Appendix A) and most likely represent a so-far unrecognized tick species [38].

### 3.2. JMTV Infection in Ticks and Tortoises

Pan-JMTV PCR and/or qRT-PCR screening of 718 tick pools (≤ 8 ticks/pool) yielded sixty-seven positive samples. Assuming only one positive tick per pool, this translates to an overall estimated individual-level JMTV prevalence of 1.5% (67/718, 95% CI 1.2–1.9). The highest individual-level JMTV prevalence was observed for *Am. sparsum* (85.7%, 6/7) and *Am.* sp. (77.1%, 27/35), both collected from tortoises sampled in Kapkuikui (Baringo county), while *Rh. appendiculatus* (0.5%,1/30) collected from sheep in Oloisinyai (Kajiado county) had the lowest prevalence (Table 1). Other positive tested tick species included *Rh. evertsi evertsi* and *Hy. truncatum* (Table 1). Jingmen tick virus infection was also observed in ticks collected from small ruminants (sheep and goats) while no JMTV was detected in ticks collected from cattle at all sampling sites (Table 1).

Out of the sixty-seven JMTV-positive samples, fifty showed 93%–94% nt identity to JMTV detected in a non-human primate from Uganda (KX377513.1) in the NS5 gene (489 bp; positions 1733–2222) as shown in Figure 2. The other seventeen samples were identified by qRT-PCR and no sequence information from these samples is available (Table 2). Due to the high prevalence of JMTV in ticks collected from tortoises in Kapkuikui, whole blood was sampled from three tortoises from the same sampling locality a month after the initial sampling and screened for JMTV infection by Pan-JMTV PCR and qRT-PCR. Two of the three whole blood samples collected from tortoises tested positive using qRT-PCR (concentration = 1.58–18.2copies/µL). From one sample, a sequence fragment could be amplified using the Pan-JMTV PCR (nucleotide sequence accession number ON158817, Table 2). The sequence showed 97%–100% nt identity to JMTV detected in ticks collected from tortoise (Appendix A) and grouped within the tick-associated JMTV-derived sequences (Figure 2).

Phylogenetic analyses based on the NS5 gene sequences showed that the sequences of this study form a monophyletic clade closely related to JMTV identified from non-human primates in Uganda. The diversity of tortoise-associated JMTV was much greater than that of sheep-associated JMTV (Figure 2). JMTV detected in *Rh. appendiculatus* collected from sheep clustered together in a distinct monophyletic clade embedded in tortoise-tick-associated viruses, suggesting a tick and/or host specific clustering pattern (Figure 2).

### 3.3. Virus Isolation and Genome Organization

Virus isolation was attempted using representative JMTV-positive samples (*n* = 42). JMTV genome copies were detected in cell culture supernatants of seven samples following two cell culture passages. No cytopathic effects (CPE) were present. Cell culture supernatants comprised *Rh. appendiculatus* (KT125) collected from sheep, from *Am. sparsum* (MT293), and from *Am.* sp. (samples MT299, MT304, MT305, MT308, and MT314) both collected from tortoise hosts. However, no virus replication was measured after the third passage in both Vero E6 and C6/36 cells. Multiple attempts to further passage the virus after two cell culture passages were not successful.

Coding complete genomes were obtained by NGS from two *Am*. sp. ticks (MT304 and MT299) inoculated into C6/36 and Vero E6 cells respectively, as well as from two *Am. sparsum* (MT290 and MT293) and three *Am*. sp. (MT297, MT308, and MT 328) homogenates. Each genome consisted of four genome segments and had a size of approximately 11 kb excluding the non-coding untranslated regions (UTRs). The genomes showed a typical segmented JMTV genome organization comprising three monocistronic segments (segments 1, 2, and 3) and two separate ORFs in segment 4 (Figure 3A). A 924-amino acid polypeptide on segment 1 was predicted to code for non-structural protein 1 (NSP1). The protein is homologous to the flavivirus NS5 protein and contains an RNA-dependent RNA polymerase (RdRp) motif (Figure 3A). The other motif present in segment 1 included S-adenosyl-L-methionine-dependent methyltransferases (Figure 3A). Further, a transmembrane domain (position 16–35) and two N-glycosylation sites at position 335 and 519 were observed (Figure 3A). Like other JMTVs, motifs A, B, and C on NSP1 were present in all sequences found in this study (Figure 3B). Analysis of RdRp genes did not reveal any unique nonsynonymous nucleotide substitutions compared to other JMTV sequences. Segment 2 was predicted to encode the structural glycoprotein VP1 and contained a potential N-glycosylation site at position 171 and 225 (Figure 3A). Segment 3 was predicted to code for the NSP2 protein, which shares functional homology with the flavivirus NS2b–NS3 complex. Further, it contained P-loop structures with nucleoside triphosphate hydrolases and RNA helicase motifs (Figure 3A). Signal peptides were also identified in NSP2. Similarly, an ATP-binding site (PGAGKTR) and DEAD-box helicase domain on NSP2 were conserved in all sequences from this study (Figure 3C). Analysis of protease/helicase genes did not reveal any unique nonsynonymous nucleotide substitutions to other JMTVs. The two separate ORFs in segment 4 were predicted to encode two polypeptide viral glycoproteins (VP) 2 and 3 (Figure 3A).

Phylogenetic analyses based on all segments showed that the Kenyan sequences formed a monophyletic clade closely related to JMTV identified in a non-human primate from Uganda and in ticks and rodents from China and ticks from Laos (Figure 4).

Notably, all viruses sequenced in this study were closely related to each other at RdRp and Protease/helicase genes with 99%–100% deduced aa identity (Appendix A).

## 4. Discussion

JMTV has recently been found in many countries, but little is known about its presence in Africa. The only report of JMTV in Africa details its detection in a non-human primate from Uganda. Here, we describe the detection of JMTV in several tick species collected from sheep, goats, and tortoises. JMTV genome copies were also detected in whole blood collected from tortoises. This represents the first record of JMTV in Kenya and in a reptilian host.

JMTV was found in tick species that were not associated with JMTV before. Positive tick species included *Rh. appendiculatus, Rh. evertsi evertsi,* and *Hy. truncatum* associated with goats and sheep, as well as *Am. sparsum, Am. nuttalli*, and *Am.* sp. associated with tortoises. JMTV has been detected in other studies in *Rh. microplus* in France, China, Brazil, and Trinidad and Tobago [11,39,40]. Other tick species that have been shown to harbor JMTV include *Am. variegatum* originating from France, *Hy. marginatum*, and *Haemaphysalis inermis* originating from Turkey, showing that JMTV infects species of the genera *Rhipicephalus, Amblyomma,*
*Haemaphysalis*, and *Hyalomma* [11,41]. The highest JMTV prevalence was detected in *Am. sparsum* (85.7%, 95% CI 50.6–99.1) and *Am*. sp. (77.1%, 95% CI 61.6–88.8) collected from tortoises in Kapkuikui. Interestingly, we did not find any tick collected from cattle positive for JMTV, although cattle from China have been shown to be infected with JMTV [1,8]. Instead, ticks infesting sheep and goats were found positive with a percentage of 1.8% (95% CI 1.2–2.5) and 0.3% (95% CI 0.1–0.5), respectively. Together with earlier findings, these results indicate that JMTV is a potential emerging tick-borne pathogen with a wide geographic distribution and infecting a wide diversity of tick species.

Integral to arbovirus disease transmission dynamics is the presence of a pathogen, a competent vector, and a susceptible animal host. In the frame of susceptible animal hosts inhabiting areas in Kenya with history of arboviral disease, tortoises are among widely distributed tick hosts. Tortoises are known to be long-lived reptiles and have low species richness, with only one or two species living in sympatry [42,43]. Common viral infections in tortoise include ranaviruses and herpesviruses [44,45,46]. However, the role of reptiles in arbovirus transmission dynamics is barely investigated despite them living near humans and domestic animals and being parasitized by ticks such as *Am. gemma*, which is also known to feed on humans and livestock and *Am. falsomarmoreum* [23,47]. In some cases, arbovirus transmission is facilitated by the presence of a reservoir/maintenance host where the virus circulates latently as in the case of Crimean–Congo hemorrhagic fever virus (CCHFV) in birds and small mammals [48,49]. Screening of tortoises’ blood for JMTV infection revealed the presence of JMTV RNA (nucleotide sequence accession number: ON158817; Table 2) indicative of JMTV presence in both the reptile and ticks from the same geographic location, revising our understanding of JMTV host range. Further, phylogenetic analyses of the NS5 gene revealed that the diversity of tortoise-associated JMTV was much greater than that of sheep-associated JMTV, suggesting that a much wider JMTV diversity may exist in tortoises (Figure 2). More sequence information would be required for a comprehensive phylogenetic analysis to investigate if (ancestral) JMTV variants may have spread to mammals including sheep. Although the feeding pattern of *Am.* sp. is not known, other tortoise-associated tick species reporting JMTV infection including *Am. sparsum* and *Am. nuttalli* have been shown to feed on livestock and humans, respectively [47,50]. Overall, the findings suggest that tortoises could become infected with JMTV. However, as the present study involved only a small number of tortoises, a focused JMTV survey including antibody screening could provide more insights into JMTV maintenance in tortoise and transmission between tortoise and ticks.

JMTV isolation attempts suggested only replication for two cell culture passages in Vero E6 and C6/36 cells. While the detection of JMTV RNA in cell culture supernatant may be an indication of virus replication in these cells, it is also possible that the JMTV RNA detections in cell culture supernatants were carryover from tick homogenates as speculated by Kobayashi et al. [51]. Earlier virus isolation attempts have shown mixed outcomes with inconsistent replication in C6/36 as well as in DH82 (canine macrophage) cells, and persistent growth in BME/CTVM23 (*Rh. microplus* embryo-derived cell line) [1,2,51]. Other jingmenviruses such as ALSV has been shown to replicate in Vero and IRE/CTVM19 (*Ixodes ricinus)* cell lines [52,53]. In the present study, JMTV was detected from seven out of forty-two cell culture supernatants only up to the second blind passages, highlighting the need to establish a more efficient JMTV isolation approach.

Seven complete-coding genomes of JMTV were sequenced and characterized. A comprehensive analysis of coding regions showed a genome structure and functional organization comparable to those of other reported JMTVs. The RNA helicase in segment 3 was comparable to flavivirus NS3 helicase responsible for viral RNA capping and synthesis [54]. The segment plays an important role in polyprotein processing and genome replication with NS3 involved in ATP-dependent RNA or dsRNA unwinding activities [1]. Further, similarities in genome segments organization, amino acid motifs, and signal peptides were noted. In addition, the monophyletic grouping of the current samples suggests that like other JMTVs, the segmented genome may be a product of a single evolutionary event [10]. However, a lot remains to be investigated on the evolutionary history of JMTV especially following its detection in tortoises, an animal known to be long living and having low species richness.

## 5. Conclusions

Our findings showed circulation of JMTV in several tick species in Kenya. Virus-positive ticks were collected from sheep, goats, and tortoises, with the latter showing the highest JMTV infection rates. Interestingly, the diversity of tortoise-associated JMTV was much greater than that of sheep-associated JMTV. Further epidemiological surveys including antibody screening of livestock and humans are necessary to appraise the potential health risk posed by JMTV in the study areas and beyond.

## Figures and Tables

**Figure 1 viruses-14-01041-f001:**
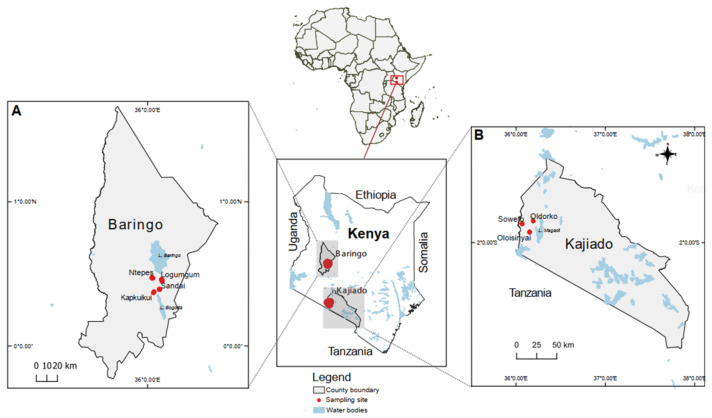
Map showing tick collection sites in Kenya. The red dots indicate sampling points. (**A**), location of Baringo county; (**B**), location of Kajiado county. The maps were created in the open source GIS software QGIS 2.12 using GPS co-ordinates and shape files derived from Natural Earth (http://www.naturalearthdata.com/, a free GIS data source, accessed on 20 October 2021) and Africa Open data (https://africaopendata.org/dataset/kenya-counties-shapefile/, license Creative Commons, accessed on 20 October 2021) [17].

**Figure 2 viruses-14-01041-f002:**
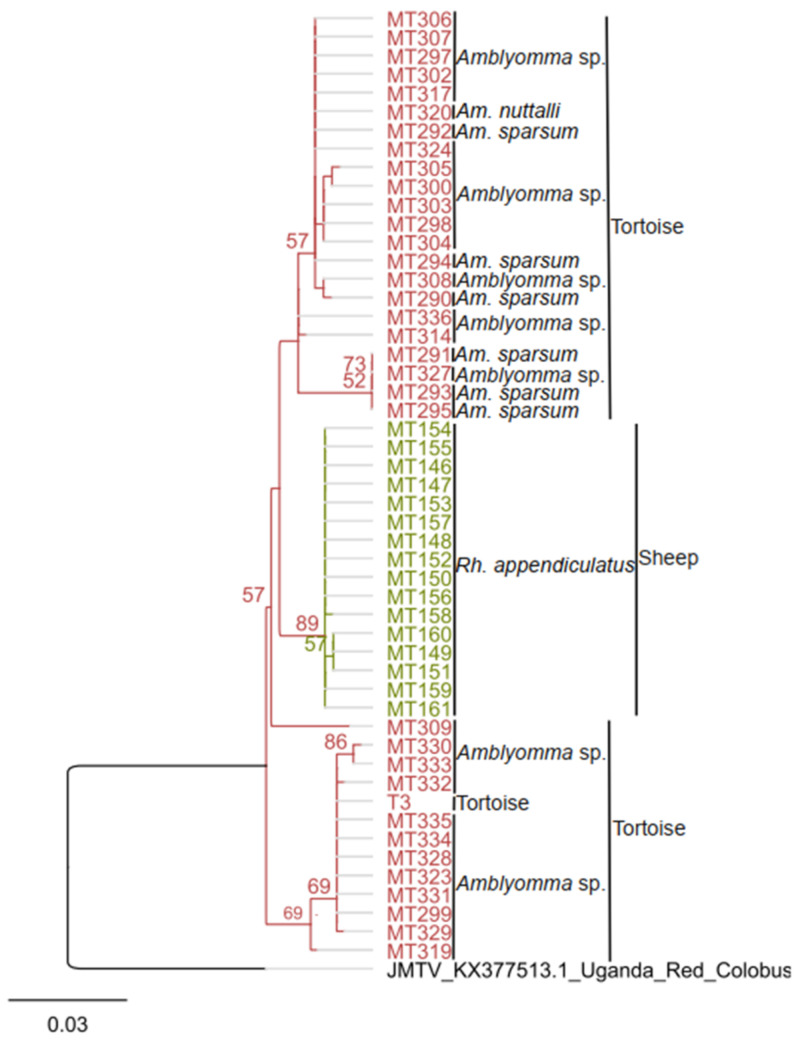
Phylogenetic relationship of detected JMTV sequences. The ML phylogenetic analysis is based on a 489-nucleotide fragment of the NS5 gene. Sequences detected in the study and a JMTV reference sequence from Uganda (KX377513.1) were aligned using MAFFT and tree inferred using PhyML v. 2.2.4 with GTR substitution models employing 1000 bootstrap replicates. Trees are midpoint rooted. Only bootstrap values exceeding 50% are shown. Species information of infected ticks and the hosts from which ticks have been collected are indicated.

**Figure 3 viruses-14-01041-f003:**
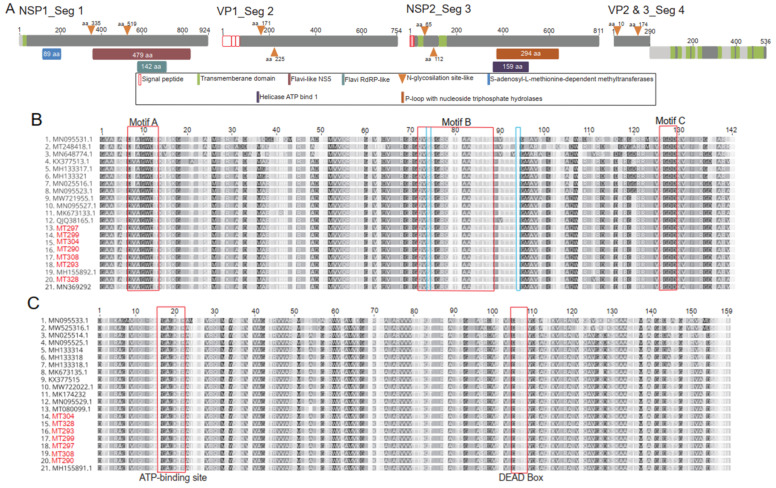
Genome structure of JMTV. (**A**) genome organization; grey regions represent non cytoplasmic domains while light grey regions show cytoplasmic domains; (**B**) conserved JMTV RNA-dependent RNA polymerase; motifs A, B, and C represent highly conserved regions, light blue boxes highlight regions of functional significance; (**C**) protease/helicase motifs of JMTV and viruses identified in the present study.

**Figure 4 viruses-14-01041-f004:**
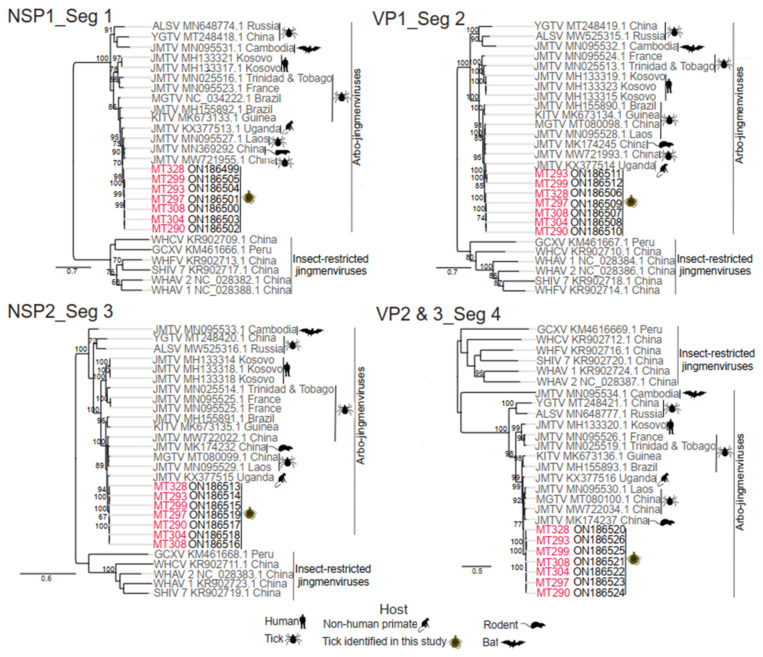
Phylogentic analyses of JMTV complete coding sequences. The maximum likelihood phylogenetic analyses were based on nucleotide sequences of segements 1, 2, 3, and 4. Host information and geographic origin of the sequences are indicated. Phylogenetic trees were inferred from MAFFT alignment using PhyML v. 2.2.4 with General-time-reversible (GTR) substitution models employing 1000 bootstrap replicates. Trees are midpoint rooted. JMTV sequences identified in this study are highlighed in red. Bootstrap support values of more than 65 are shown.

**Table 1 viruses-14-01041-t001:** Estimated individual-level JMTV prevalence in ticks collected from different hosts.

County	Sampling Site	Tick Species	Goats (*n*)	Sheep (*n*)	Tortoises (*n*)	Cattle (*n*)
Kajiado	Oloisinyai	*Rh. appendiculatus*	0 (0/4)	0.5 (1/30)	0 (0/0)	0 (0/5)
Baringo	Ntepes	*Rh. appendiculatus*	0 (0/84)	4.4 (16/55)	0 (0/0)	0 (0/0)
	Sandai	*Rh. evertsi evertsi*	0 (0/0)	42.3 (1/2)	0 (0/0)	0 (0/2)
		*Rh. appendiculatus*	0.5 (1/28)	0 (0/7)	0 (0/0)	0 (0/20)
		*Hy. truncatum*	0 (0/2)	29.3 (1/2)	0 (0/0)	0 (0/3)
	Logumgum	*Hy. truncatum*	15.5 (1/3)	0 (0/2)	0 (0/0)	0 (0/0)
		*Rh. appendiculatus*	1.5 (4/35)	3.3 (7/30)	0 (0/0)	0 (0/0)
	Kapkuikui	*Am.* sp.	0 (0/0)	0 (0/0)	77.1 (27/35)	0 (0/0)
		*Am. sparsum*	0 (0/0)	0 (0/0)	85.7 (6/7)	0 (0/0)
		*Am. nuttalli*	0 (0/0)	0 (0/0)	40.0 (2/5)	0 (0/0)
Total (*n*)			0.3 (6/156)	1.8 (26/128)	74.5 (35/47)	0 (0/30)

*n*: proportion of positive pools.

**Table 2 viruses-14-01041-t002:** Characteristics of JMTV-positive ticks and tortoises in Baringo and Kajiado counties, Kenya.

Code	County	Sampling Site	Time of Sample Collection	Species	Pool Size	Host	Sequence Length (nt)	GenBank Accession No
MT146	Baringo	Ntepes	10 August 2019	*Rh. appendiculatus*	8♀	Sheep	594	ON158858
MT147		Ntepes	10 August 2019	*Rh. appendiculatus*	8♀	Sheep	595	ON158857
MT148		Ntepes	10 August 2019	*Rh. appendiculatus*	8♀	Sheep	602	ON158854
MT149		Ntepes	10 August 2019	*Rh. appendiculatus*	8♀	Sheep	594	ON158864
MT150		Ntepes	10 August 2019	*Rh. appendiculatus*	8♀	Sheep	591	ON158852
MT151		Ntepes	10 August 2019	*Rh. appendiculatus*	8♀	Sheep	589	ON158862
MT152		Ntepes	10 August 2019	*Rh. appendiculatus*	8♀	Sheep	593	ON158853
MT153		Ntepes	10 August 2019	*Rh. appendiculatus*	8♀	Sheep	586	ON158856
MT154		Ntepes	10 August 2019	*Rh. appendiculatus*	8♀	Sheep	604	ON158860
MT155		Ntepes	10 August 2019	*Rh. appendiculatus*	8♀	Sheep	588	ON158859
MT156		Ntepes	10 August 2019	*Rh. appendiculatus*	8♀	Sheep	591	ON158851
MT157		Ntepes	10 August 2019	*Rh. appendiculatus*	8♀	Sheep	591	ON158855
MT158		Ntepes	10 August 2019	*Rh. appendiculatus*	8♀	Sheep	590	ON158865
MT159		Ntepes	10 August 2019	*Rh. appendiculatus*	8♀	Sheep	595	ON158861
MT160		Ntepes	10 August 2019	*Rh. appendiculatus*	8♀	Sheep	596	ON158863
MT161		Ntepes	10 August 2019	*Rh. appendiculatus*	8♀	Sheep	592	ON158850
MT290		Kapkuikui	10 August 2019	*Am. sparsum*	1♂	Tortoise	577	ON158844
MT291		Kapkuikui	10 August 2019	*Am. sparsum*	1♂	Tortoise	578	ON158847
MT292		Kapkuikui	10 August 2019	*Am. sparsum*	1♂	Tortoise	580	ON158831
MT293		Kapkuikui	10 August 2019	*Am. sparsum*	1♂	Tortoise	561	ON158846
MT294		Kapkuikui	10 August 2019	*Am. sparsum*	1♂	Tortoise	590	ON158837
MT295		Kapkuikui	10 August 2019	*Am. sparsum*	1♂	Tortoise	539	ON158845
MT297		Kapkuikui	10 August 2019	*Amblyomma* sp.	1♂	Tortoise	559	ON158835
MT298		Kapkuikui	10 August 2019	*Amblyomma* sp.	1♂	Tortoise	538	ON158841
MT299		Kapkuikui	10 August 2019	*Amblyomma* sp.	1♂	Tortoise	532	ON158822
MT300		Kapkuikui	10 August 2019	*Amblyomma* sp.	1♂	Tortoise	531	ON158842
MT302		Kapkuikui	12 August 2019	*Amblyomma* sp.	1♂	Tortoise	580	ON158834
MT303		Kapkuikui	12 August 2019	*Amblyomma* sp.	1♂	Tortoise	559	ON158840
MT304		Kapkuikui	12 August 2019	*Amblyomma* sp.	1♂	Tortoise	559	ON158839
MT305		Kapkuikui	12 August 2019	*Amblyomma* sp.	1♂	Tortoise	489	ON158843
MT306		Kapkuikui	12 August 2019	*Amblyomma* sp.	1♂	Tortoise	524	ON158836
MT307		Kapkuikui	12 August 2019	*Amblyomma* sp.	1♂	Tortoise	516	ON158849
MT308		Kapkuikui	12 August 2019	*Amblyomma* sp.	1♂	Tortoise	520	ON158838
MT309		Kapkuikui	12 August 2019	*Am. nuttalli*	1♀	Tortoise	562	ON158866
MT314		Kapkuikui	12 August 2019	*Amblyomma* sp.	1♂	Tortoise	577	ON158829
MT317		Kapkuikui	12 August 2019	*Amblyomma* sp.	1♂	Tortoise	578	ON158833
MT319		Kapkuikui	12 August 2019	*Amblyomma* sp.	1♂	Tortoise	562	ON158827
MT320		Kapkuikui	13 August 2019	*Am. nuttalli*	1♂	Tortoise	603	ON158832
MT323		Kapkuikui	13 August 2019	*Amblyomma* sp.	1♂	Tortoise	588	ON158821
MT324		Kapkuikui	13August 2019	*Amblyomma* sp.	1♂	Tortoise	584	ON158830
MT327		Kapkuikui	14 August 2019	*Amblyomma* sp.	1♂	Tortoise	531	ON158848
MT328		Kapkuikui	14 August 2019	*Amblyomma* sp.	1♂	Tortoise	520	ON158820
MT329		Kapkuikui	14 August 2019	*Amblyomma* sp.	1♂	Tortoise	531	ON158867
MT330		Kapkuikui	15 August 2019	*Amblyomma* sp.	1♂	Tortoise	514	ON158826
MT331		Kapkuikui	15 August 2019	*Amblyomma* sp.	1♂	Tortoise	519	ON158823
MT332		Kapkuikui	15 August 2019	*Amblyomma* sp.	1♂	Tortoise	565	ON158824
MT333		Kapkuikui	15 August 2019	*Amblyomma* sp.	1♂	Tortoise	568	ON158825
MT334		Kapkuikui	15 August 2019	*Amblyomma* sp.	1♂	Tortoise	575	ON158819
MT335		Kapkuikui	15 August 2019	*Amblyomma* sp.	1♂	Tortoise	575	ON158818
MT336		Kapkuikui	15 August 2019	*Amblyomma* sp.	1♂	Tortoise	571	ON158828
MT4		Logumgum	15 August 2019	*Rh. appendiculatus*	8♂	Sheep	‡	‡
MT8		Logumgum	15 August 2019	*Rh. appendiculatus*	8♂	Sheep	‡	‡
MT19		Logumgum	15 August 2019	*Rh. appendiculatus*	8♀	Sheep	‡	‡
MT23		Logumgum	15 August 2019	*Rh. appendiculatus*	8♀	Sheep	‡	‡
MT26		Logumgum	15 August 2019	*Rh. appendiculatus*	8♀	Sheep	‡	‡
MT29		Logumgum	15 August 2019	*Rh. appendiculatus*	8♀	Sheep	‡	‡
MT31		Logumgum	15 August 2019	*Rh. appendiculatus*	8♀	Sheep	‡	‡
MT42		Logumgum	15 August 2019	*Rh. appendiculatus*	8♀	Goat	‡	‡
MT54		Logumgum	15 August 2019	*Rh. appendiculatus*	8♂	Goat	‡	‡
MT55		Logumgum	15 August 2019	*Rh. appendiculatus*	8♂	Goat	‡	‡
MT61		Logumgum	15 August 2019	*Hy. truncatum*	2♂	Goat	‡	‡
MT62		Logumgum	15 August 2019	*Rh. appendiculatus*	8♀	Goat	‡	‡
MT101		Sandai	15 August 2019	*Rh. appendiculatus*	8♂	Goat	‡	‡
MT136		Sandai	15 August 2019	*Hy. truncatum*	2♂	Sheep	‡	‡
MT144		Sandai	15 August 2019	*Rh. evertsi evertsi*	2♂	Sheep	‡	‡
MT313		Kapkuikui	12 Auguast 2019	*Amblyomma sp.*	1♂	Tortoise	‡	‡
T3 *		Kapkuikui	29 September 2019		1♀		596	ON158817
T2 *		Kapkuikui	29 September 2019		1♀		‡	‡
KT125	Kajiado	Oloisinyai	19 July 2020	*Rh. appendiculatus*	8♂	Sheep	‡	‡

♀: female; ♂: male; nt: nucleotides; *: sequence was derived from vertebrate; ‡: positive by qPCR only, no sequence information available.

## Data Availability

Sequences generated were deposited to GenBank under accession numbers ON158817–ON158867, ON186499–ON186526, ON220154–ON220159 and ON212401–ON212405. Other data presented in the study are available in the article and as Appendix A.

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
