# Peer review of "Jingmen Tick Virus in Ticks from Kenya"

_viruses, 2022, doi:10.3390/v14051041_

Round 1

Reviewer 1 Report

Thank you for the production of your article, which indictes circulation of a Jingmen tick virus in a novel region.  The focus on JMTV seems less hypothesis driven and more an exploration, of which I feel too much emphasis has been placed on the tortoise samples obtained (potentially from the same location). The finding of this virus in Kenya however is very interesting, and the manuscript is generally well-written.  I have minor comments, and suggestions for improvement as noted below:

Overall, reduce the emphasis on tortoises (unless you can sample additional tortoises from more than that 1 site), and broaden the focus to the virus itself and the different hosts in general.  Consider taking tortoise out of the title.

Abstract:   Provide detail here as to the species of tick tested.   State the number of ticks [i.e., only n=47 ~ 1%] that came from tortoises.  In fact, I would like to see N for each of the animal taxa.

Line 44) Pluralize the word ‘primateS’

50-54) I worked out what is meant here, but it could be written more clearly

65) Remove ‘The’ at the start of the sentence

68) Why would you assume that the aetiology is a tick-borne virus or JTMV, as opposed to a mosquito-borne arbovirus or a non-viral pathogen?

87) How were the ticks collected/removed from respective animals?

94)  Was the life-stage of ticks considered?  (Recommend that it is)

Figure 1) Amend/reduce font size of legend

Consider including the English name ‘Leopard tortoise’, within descriptions.

105) How many tortoises were sampled??   Also, where were these located – free-ranging wildlife, or captive?   Which locations within Baringo county?  Only Kapkuikui??

[Why] Were all the tortoises collected from the same locality within Kapkuikui?  Given their interactions, that could greatly influence why you found so many positive.  Suggest broadening tortoise sampling over a wider area; the same way that goats, sheep and cattle were sampled.

106) How much blood was taken from an individual?

Section 2.4:   Please modify the Title of this section to better reflect the described methods (i.e., the ticks are homogenized also for initial stages of pathogen detection). 

Make it clearer also why you are using molecular identification methods / i.e., was morphological ID not reliable?  For what species was this used, if not all? 

150) Sample being the supernatant of tick homogenates?

Section 2.6: Check over again for succinctness.  List manufacturer, city, country for each reagent.   155) put Gibco afterwards; 156) FCS details given above so not needed again.

167-68) Needs further explanation

182-186) Cite the relevant software(s)

189) Provide Genbank ref earlier when first mentioned.

Section 3.1:  The latter half of this section could be written more concisely with a clearer flow.

S1:  Nice figure

204)  So, you think these are novel tick species from the tortoises?  This is really interesting, but not given much attention compared to the virus sequence data. Given that most of the JMTV-positive samples came from these unknown Amblyomma specimens, suggest focusing more on determining what they are.  (Although given this is the journal Viruses, I’ll let the editor call that one). 

Table 2:  I’m not clear from your methodology why the sequence lengths differ

365-376) & 406-408) I’m not sure we agree with this, so would tone this conclusion down

Discussion/Conclusion & 370) – Finding viral RNA in a vertebrate doesn’t necessarily mean it could act as an amplifying/reservoir host to transmit the virus onwards, so please be careful not to jump to implying that tortoises are a new competent risk for arboviral disease transmission; the tortoises simply gained an infection (as did goats/sheep).

Table S2 – Write each genus out in full

Author Response

Reviewer 1:

Thank you for the production of your article, which indicates circulation of a Jingmen tick virus in a novel region. The focus on JMTV seems less hypothesis driven and more an exploration, of which I feel too much emphasis has been placed on the tortoise samples obtained (potentially from the same location). The finding of this virus in Kenya however is very interesting, and the manuscript is generally well-written. I have minor comments, and suggestions for improvement as noted below:

  1. Overall, reduce the emphasis on tortoises (unless you can sample additional tortoises from more than that 1 site), and broaden the focus to the virus itself and the different hosts in general. Consider taking tortoise out of the title.

Re: Thank you for highlighting this. Tortoise has been taken out from the title, and the title now reads “Jingmen tick virus in ticks from Kenya”. In addition, the part describing the tortoise-associated JMTV sequences was rephrased in the abstract and now reads: “Intragenetic distance of JMTV sequences originating from tortoise-associated ticks was greater than that of sheep-associated ticks.” L30-31.

  1. Abstract:   Provide detail here as to the species of tick tested. State the number of ticks [i.e., only n=47 ~ 1%] that came from tortoises. In fact, I would like to see N for each of the animal taxa.

Re: Thank you. This has been revised accordingly and the sentence now reads: “…cattle (n=859, 18.88%), goats (n=2070, 45.49%), sheep (n=1574, 34.59%), and free-ranging tortoises (Leopard tortoise, Stigmochelys pardalis) (n=47, 1.03%).” L24-26

  1. Line 47) Pluralize the word ‘primateS’

Re: Thank you. This has been revised and now reads “primates”. L47

  1. 50-54) I worked out what is meant here, but it could be written more clearly.

Re: Thank you for raising this issue. This has been revised and now reads: “Jingmenviruses form two phylogenetic groups. The viruses of one group are transmitted by blood-feeding arthropods and are vertebrate pathogenic (arbo-Jingmenviruses) such as JMTV, Mogiana tick virus (MGTV) and Yanggou tick virus (YGTV) [2,10–12]. The second group contains viruses that only infect arthropods (arthropod-specific jingmenviruses) exemplified by Guaico Culex virus (GCXV), Wuhan aphid virus (WHAV) and Shuangao insect virus (SHIV) group [8,10,12]. L52-57

  1. 65) Remove ‘The’ at the start of the sentence

Re: This has been revised accordingly.

  1. 68) Why would you assume that the aetiology is a tick-borne virus or JMTV, as opposed to a mosquito-borne arbovirus or a non-viral pathogen?

Re: Thank you for raising the issue. We do not assume that the aetiology of febrile human illness is a tick-borne virus or JMTV. The findings reported here are part of an active arbovirus survey to identify circulating and novel agents that could be the cause of undiagnosed febrile human illness in the study area. We moved this sentence (“Both areas have a history of undiagnosed febrile human illness, with poorly described aetiologies.”) to the study site description in the methods section 2.2.

  1. 87) How were the ticks collected/removed from respective animals?

Re: Thank you for picking up the missing information. Before ticks’ collection, the individual livestock (cattle, goats and sheep) was restrained to allow manual picking of attached, adult ticks from the animal’s skin. This sentence has now been added to the methods section part 2.2.

  1. 94) Was the life-stage of ticks considered?  (Recommend that it is)

Re: Thank you for your concern. Yes, we sampled adult ticks. 23). The word “Adult” ticks has been added to the text (line 80 and 89).

  1. Figure 1) Amend/reduce font size of legend.

Re: The legend font size has been reduced as suggested and a revised figure 1 uploaded.

  1. Consider including the English name ‘Leopard tortoise’, within descriptions. L25 & 90

Re: Thank you for highlighting this. The English name has been added within descriptions.

  1. 105) How many tortoises were sampled??   Also, where were these located – free-ranging wildlife, or captive?   Which locations within Baringo county?  Only Kapkuikui??

Re: Thank you for the questions. All requested information has been added to the text. Ticks were sampled from randomly available free-ranging tortoises in Kapkuikui, Baringo county. However, blood was obtained from three individual tortoises (lines 25;109-110; 265-266).

  1. [Why] Were all the tortoises collected from the same locality within Kapkuikui?  Given their interactions, that could greatly influence why you found so many positive. Suggest broadening tortoise sampling over a wider area; the same way that goats, sheep and cattle were sampled.

Re: The reviewer has a valid question and gives a plausible reason for a change in sample design. Tortoises were collected from the same locality within Kapkuikui due to the high prevalence of JMTV in ticks collected from tortoises in Kapkuikui. The other sampling sites in the present study are not inhabited by tortoises and could therefore not be included. Future surveys will focus on other areas inhabited by tortoises.

  1. 106) How much blood was taken from an individual?

Re: We added the requested information to the text (lines 113-116). It now reads: “About 0.5 ml of whole blood was collected aseptically using 2mL BD Vacutainer with EDTA (Becton Dickinson, USA) or approximately 50 µl of whole blood spotted on FTA Classic Card Whatman (Merck KGaA; Darmstadt, Germany).”

  1. Section 2.4: Please modify the Title of this section to better reflect the described methods (i.e., the ticks are homogenized also for initial stages of pathogen detection). 

Re: Thank you. The title has been revised as suggested and now reads: “Tick Homogenization and Molecular Identification” L120

  1. Make it clearer also why you are using molecular identification methods / i.e., was morphological ID not reliable?  For what species was this used, if not all? 

Re: Thank you for raising the issue. Molecular identification methods were used for those Amblyoma ticks that could not be identified to species level based on morphological identification keys. This has been added to the text (lines 126-128). “…. genomic DNA was extracted from the pellet of Amblyomma ticks that could not be identified to species level by morphology…..”.

  1. 150) Sample being the supernatant of tick homogenates?

Re: Thank you. Yes, this is correct. “supernatant of tick homogenate” has been added to the text: “In the filtered approach, 100 µL of each sample (supernatant of tick homogenate) was passed through 0.45 µm sterile membrane filters …”. L157

  1. Section 2.6: Check over again for succinctness.  List manufacturer, city, country for each reagent.   155) put Gibco afterwards; 156) FCS details given above so not needed again.

Re: Thank you for the bringing this to our attention. This has been revised and the text now reads: “… (FCS; Gibco, Thermo Fisher Scientific, Darmstadt, Germany).” In line 164 manufacturer, city, country has been deleted.

  1. 167-68) Needs further explanation

Re: The sentence has been revised and now reads… “A quantitative TaqMan Real-time PCR (qRT-PCR) using JMTV-specific primers and probe (Supplementary Table S1) was established to measure the amount of viral genome copies in the collected cell culture supernatants”.

  1. 182-186) Cite the relevant software(s)

Re: Thank you. The reference (http://cab.spbu.ru/software/spades/) for Spades v3.11.1 has been provided. In addition, Geneious and BLAST references are provided in L195.

  1. 189) Provide Genbank ref earlier when first mentioned.

Re: Thank you. The GenBank ref have been provided. “…. ON158817–ON158867, ON186499–ON186526, ON220154–ON220159 and ON212401– ON212405…..” L218-224

  1. Section 3.1:  The latter half of this section could be written more concisely with a clearer flow.

Re: Thank you. This part has been revised and now reads: "Thirty-five Amblyomma ticks collected from tortoises could not be identified to species level by morphology nor by sequencing of ITS2 and 16S rRNA gene fragments. These ticks are denoted as Am. sp., henceforth. The generated sequences showed 99-100% pairwise nucleotide identities within the 16S rRNA gene fragment, and 98-100% pairwise nucleotide identities within the ITS2 fragment. Amplification of the CO1 gene was not successful. Phylogenetic analysis of ITS2 fragment sequences (1,008 bp nucleotides in length) revealed that the Am. Sp. ticks are most closely related to Am. marmoreum and Am. moculosum showing pairwise nucleotide identities of 93-94% to both species (Supplementary Figure S1). Analysis of the 16S rRNA gene fragment (402 bp nucleotides in length) showed a similar relationship of the collected Am. sp. ticks to Am. marmoreum (93-94% nt similarity). These data suggest that the collected Am. sp. ticks are distinct from formally recognized Amblyomma tick species for which homologous 16S rRNA and ITS2 data are available (Supplementary Figure S1) and most likely represent a so far unrecognized tick species [39].” L234-247

  1. S1:  Nice figure

Re: Thank you.

  1. 204) So, you think these are novel tick species from the tortoises?  This is really interesting, but not given much attention compared to the virus sequence data. Given that most of the JMTV-positive samples came from these unknown Amblyomma specimens, suggest focusing more on determining what they are.  (Although given this is the journal Viruses, I’ll let the editor call that one). 

Re: Thank you for the notification. Unfortunately, these tick species could not be identified by taxonomic keys nor by sequencing of ITS2 and 16S rRNA gene fragments. It is very likely that these tick species may indeed represent novel species. However, further studies need to focus on the characterization of this tick species.

  1. Table 2:  I’m not clear from your methodology why the sequence lengths differ.

Re: Thank you for bringing up the issue. Some samples were found positive by qPCR but amplification of sequence fragments was not successful. This information is now added as a footnote to the table. Primer sequences and sequence lengths are provided in Table S1.

  1. 365-376) & 406-408) I’m not sure we agree with this, so would tone this conclusion down.

Re: Thank you for the suggestion. We toned this down and the text now reads: “The diversity of tortoise-associated JMTV was much greater than that of sheep-associated JMTV (Figure 2). JMTV detected in Rh. appendiculatus collected from sheep clustered together in a distinct monophyletic clade embedded in tortoise-tick-associated viruses suggesting a tick and / or host specific clustering pattern (Figure 2).”

  1. Discussion/Conclusion & 370) – Finding viral RNA in a vertebrate doesn’t necessarily mean it could act as an amplifying/reservoir host to transmit the virus onwards, so please be careful not to jump implying that tortoises are a new competent risk for arboviral disease transmission; the tortoises simply gained an infection (as did goats/sheep).

Re: Thank you for the concern. The reviewer is right. As already discussed above the parts addressing the finding of the virus in tortoise and in ticks feeding on tortoises have been phrased more carefully now. The respective discussion part now reads: “Further, phylogenetic analyses of the NS5 gene revealed the diversity of tortoise-associated JMTV was much greater than that of sheep-associated JMTV suggesting that a much wider JMTV diversity may exist in tortoises (Figure 2). More sequence information would be required for a comprehensive phylogenetic analysis to investigate if (ancestral) JMTV variants may have spread to mammals including sheep. Although the feeding pattern of Am. sp. is not known, other tortoise-associated tick species reporting JMTV infection including Am. sparsum and Am. nuttalli have been shown to feed on livestock and human, respectively [48,51]. Overall, the findings suggest that tortoises could become infected with JMTV. To the best of our knowledge this is the first detection of JMTV in reptiles expanding the vertebrate hosts’ range of the virus. However, as the present study involved only a small number of tortoises, a focused JMTV survey including antibody screening could provide more insights into JMTV maintenance in tortoise and transmission between tortoise and ticks.” L373-383

The respective conclusion part now reads: “Our findings show circulation of JMTV in several tick species in Kenya. Virus positive ticks were collected from sheep, goat and tortoises, with the latter showing the highest JMTV infection rates. Interestingly, the diversity of tortoise-associated JMTV was much greater than that of sheep-associated JMTV. Further epidemiological surveys including antibody screening of livestock and humans are necessary to appraise the potential health risk posed by JMTV in the study areas and beyond.” L409-412

Regarding the passage 365-376, please see above point 25.

  1. Table S2 – Write each genus out in full

Re: Thank you. This has been implemented throughout the table.

Reviewer 2 Report

The authors screened for the Jingmen tick virus (JMTV) in adult ticks collected from cattle, goats, sheep, and tortoises in two Kenyan areas. They found viral RNA in sheep-, goat- and tortoise-associated ticks, and also in tortoise's blood. The authors used several molecular and bioinformatic techniques to understand the phylogenetic relationships between the viral sequences with the host. The authors tried to isolate JMTV in Vero E6 cells and C6/36 and they also were able to generate seven complete JMTV genomes using NGS from the RNA isolation of tick-associated tortoises.

This work has important implications for the viral surveillance of an unstudied virus such as JMTV, providing the literature and the local community with more information that might link JMTV with undiagnosed human febrile illnesses. A next step for this study would be screening human samples for JMTV and associating it with human interactions with the animal host.

Unfortunately, it was not possible to isolate JMTV in tissue culture conditions, even though viral isolation of RNA viruses from the environment has been proved to be difficult specially when there is not enough information about the viral replication conditions. Even with reverse genetics techniques, which has been proved to be very efficient for RNA viruses, it might present a challenge for viruses with segmented genomes.

My suggestion is that the authors may clarify what exactly they mean with "inconsistent replication". Were the viral RNA titers measured by RT-qPCR after several passages? If the authors found log-increased RNA levels, then this is an indicative of successful viral replication. If not, the cell type used is probably the not the best for viral isolation.

Author Response

Reviewer 2:

The authors screened for the Jingmen tick virus (JMTV) in adult ticks collected from cattle, goats, sheep, and tortoises in two Kenyan areas. They found viral RNA in sheep-, goat- and tortoise-associated ticks, and also in tortoise’s blood. The authors used several molecular and bioinformatic techniques to understand the phylogenetic relationships between the viral sequences with the host. The authors tried to isolate JMTV in Vero E6 cells and C6/36 and they also were able to generate seven complete JMTV genomes using NGS from the RNA isolation of tick-associated tortoises.

This work has important implications for the viral surveillance of an unstudied virus such as JMTV, providing the literature and the local community with more information that might link JMTV with undiagnosed human febrile illnesses. A next step for this study would be screening human samples for JMTV and associating it with human interactions with the animal host.

Unfortunately, it was not possible to isolate JMTV in tissue culture conditions, even though viral isolation of RNA viruses from the environment has been proved to be difficult specially when there is not enough information about the viral replication conditions. Even with reverse genetics techniques, which has been proved to be very efficient for RNA viruses, it might present a challenge for viruses with segmented genomes.

My suggestion is that the authors may clarify what exactly they mean with “inconsistent replication”. Were the viral RNA titers measured by RT-qPCR after several passages? If the authors found log-increased RNA levels, then this is an indicative of successful viral replication. If not, the cell type used is probably the not the best for viral isolation.

Re: Thank you. The reviewer is right and has a valid suggestion. 408-410) The viral RNA titres were measured by RT-qPCR, however, viral RNA could only be detected up to the second cell culture passage. We added the following information to the text: lines 294-295) “no virus replication was measured after the third passage in both Vero E6 and C6/36 cells.” Line 394) We highlight the need to establish a more efficient JMTV isolation approach.